# Shear-induced damped oscillations in an epithelium depend on actomyosin contraction and E-cadherin cell adhesion

**Ehsan Sadeghipour[1,2†], Miguel A Garcia[3†], William James Nelson[3,4], Beth L Pruitt[1,2,4,5,6,7,8]***

[1]Department of Bioengineering, Stanford University, Stanford, United States; [2]Department of Mechanical Engineering, Stanford University, Stanford, United States; [3]Department of Biology, Stanford University, Stanford, United States; [4]Department of Molecular and Cellular Physiology, Stanford University, Stanford, United States; [5]The Stanford Cardiovascular Institute, Stanford University, Stanford, United States; [6]Mechanical Engineering, University of California, Santa Barbara, United States; [7]Biomolecular Science and Engineering, University of California, Santa Barbara, United States; [8]Cellular and Developmental Biology, University of California, Santa Barbara, United States

**Abstract** Shear forces between cells occur during global changes in multicellular organization during morphogenesis and tissue growth, yet how cells sense shear forces and propagate a response across a tissue is unknown. We found that applying exogenous shear at the midline of an epithelium induced a local, short-term deformation near the shear plane, and a long-term collective oscillatory movement across the epithelium that spread from the shear-plane and gradually dampened. Inhibiting actomyosin contraction or E-cadherin *trans*-cell adhesion blocked oscillations, whereas stabilizing actin filaments prolonged oscillations. Combining these data with a model of epithelium mechanics supports a mechanism involving the generation of a shear-induced mechanical event at the shear plane which is then relayed across the epithelium by actomyosin contraction linked through E-cadherin. This causes an imbalance of forces in the epithelium, which is gradually dissipated through oscillatory cell movements and actin filament turnover to restore the force balance across the epithelium.
DOI: https://doi.org/10.7554/eLife.39640.001

***For correspondence:**
blp@ucsb.edu

[†]These authors contributed equally to this work

**Competing interests:** The authors declare that no competing interests exist.

## Introduction

Mechanical forces play important roles at the single-cell and multicellular levels in tissue biology (*Farge, 2011*; *Hoffman et al., 2011*), including in the regulation of cell shape and movement (*Aegerter-Wilmsen et al., 2012*; *Legoff et al., 2013*; *Mao et al., 2013*; *Clément et al., 2017*), cell-cycle progression (*Campinho et al., 2013*; *Streichan et al., 2014*), cell-division orientation (*di Pietro et al., 2016*; *Gloerich et al., 2017*), and gene expression (*Aragona et al., 2013*; *Benham-Pyle et al., 2015*). In general, these effects have been studied by observing the generation of cellular traction forces, or by applying external tensile forces to whole cell populations (*Mukundan et al., 2013*; *Simmons et al., 2011*; *Tambe et al., 2011*; *Zaritsky et al., 2015*; *Micoulet et al., 2005*). While a few studies have estimated the shear forces between cells from observations of cellular tractions (*Tambe et al., 2011*; *Zaritsky et al., 2015*), none have perturbed a tissue with direct application of shear loading.

In-plane shear forces exist within any solid material under load, such as between groups of cells in tissues under applied or intrinsic forces. These forces are distinct from shear stresses applied by laminar liquid flow on top of cells that mimic the vasculature. Tissues spontaneously generate internal shear forces to maintain an external force balance (*Prost et al., 2015*), and biopolymer networks have been observed to pull inward when sheared on a rheometer (*Janmey et al., 2007*).

In-plane shear forces are thought to affect the collective behavior of migrating cells in epithelial tissues (*Tambe et al., 2011*; *Zaritsky et al., 2015*). For example, the ridges that appear in the epithelial tissue forming the pupal *Drosophila* wing have been attributed to shear forces arising during development (*Etournay et al., 2015*). In addition, shear forces between migrating cells of the prechordal plate in the zebrafish embryo and cells of the neurectoderm determine the position of the neural anlage (*Smutny et al., 2017*). These studies suggest that local shear forces between groups of cells are important contributors to global effects in tissue motility and organ patterning. However, how local in-plane shear forces are spread throughout a tissue, which is important for understanding collective tissue behavior, is not understood in part because of the difficulty in applying direct and localized in-plane shear within a tissue.

In order to close this gap, here we examined epithelial mechanics after we applied in-plane shear with a novel silicon device. We determined that in-plane shear produces local deformations that are propagated into a global migratory response that distributes and dissipates forces through oscillations. Confined epithelia, similar to embryos or tumors, have been shown to oscillate (*Deforet et al., 2014*; *Kocgozlu et al., 2016*), but the mechanism driving these oscillations is unknown. Such oscillatory behavior may be important as an intrinsic collective cellular process that follows a shear-induced force imbalance, enabling the probing and maintenance of tension homeostasis within a developing tissue.

## Results

We designed and deployed a new silicon device (adapted from [*Mukundan and Pruitt, 2009*]) to apply localized shear to an epithelium while simultaneously observing cell movements and measuring forces across the epithelium (*Figure 1A–C*; Materials and methods). We fabricated devices from single crystal silicon-on-insulator wafers because silicon does not change elasticity over time (*Hopcroft et al., 2010*). The device consisted of two parallel 1000 µm x 250 µm suspended planks, one for force actuation and the other for force sensing. Moving the actuation plank applied 100 µm of shear (resulting in about one radian average cellular shear strain in cells near the mid-plane) at the midline of a Madin-Darby Canine Kidney (MDCK) epithelial cell monolayer cultured across the surface of both planks (*Figure 1A*; Materials and methods). We generated kymographs of cell movements using Particle Image Velocimetry (PIV) (*Figure 1B*), from which we mapped cell movements in the x- and y-directions relative to shear (*Figure 1C*; Materials and methods). We calculated force across the monolayer from the displacement of the sensing spring ($k_s$ = 0.93 N/m) (*Figure 1—figure supplement 1*).

Following the application of shear, the epithelium did not rupture or tear (*Figure 1—figure supplement 2*), nor was there significant extrusion of dead cells (*Video 1*). Throughout the response, cells within the epithelium retained their nearest neighbors, and new cell adhesions were formed only between daughter cells and their neighbors following division (*Figure 1—figure supplement 3*; *Video 2*).

The MDCK epithelium had a collective response to the application of in-plane shear (*Figure 1D–G*). Cells throughout the epithelium began a collective wave of inward y-direction movement toward the shear-plane, which started at the shear-plane and propagated to the opposite edge of the plank within 45 min at a rate 10x faster than individual cell velocities (290 vs. 30 µm/h; *Figure 1D,E*). This inward movement was expected because biopolymer networks develop an inward normal stress under applied shear (*Janmey et al., 2007*). In our live epithelium, cells reversed y-direction movements at 7, 11, and 15 hr after shear with decreasing velocity magnitude: they behaved as a damped oscillator (*Figure 1D,E*), with cells furthest from the shear-plane exhibiting the most movement in the y-direction (*Figure 1E*). Reversals in y-direction movement propagated from the shear plane outward at 80–100 µm/h.

Cells adjacent to the shear-plane (2–3 cell layers, <50 µm) were deformed, and moved in the x-direction opposite the applied shear (*Figure 1F,G*). These x-direction movements diminished

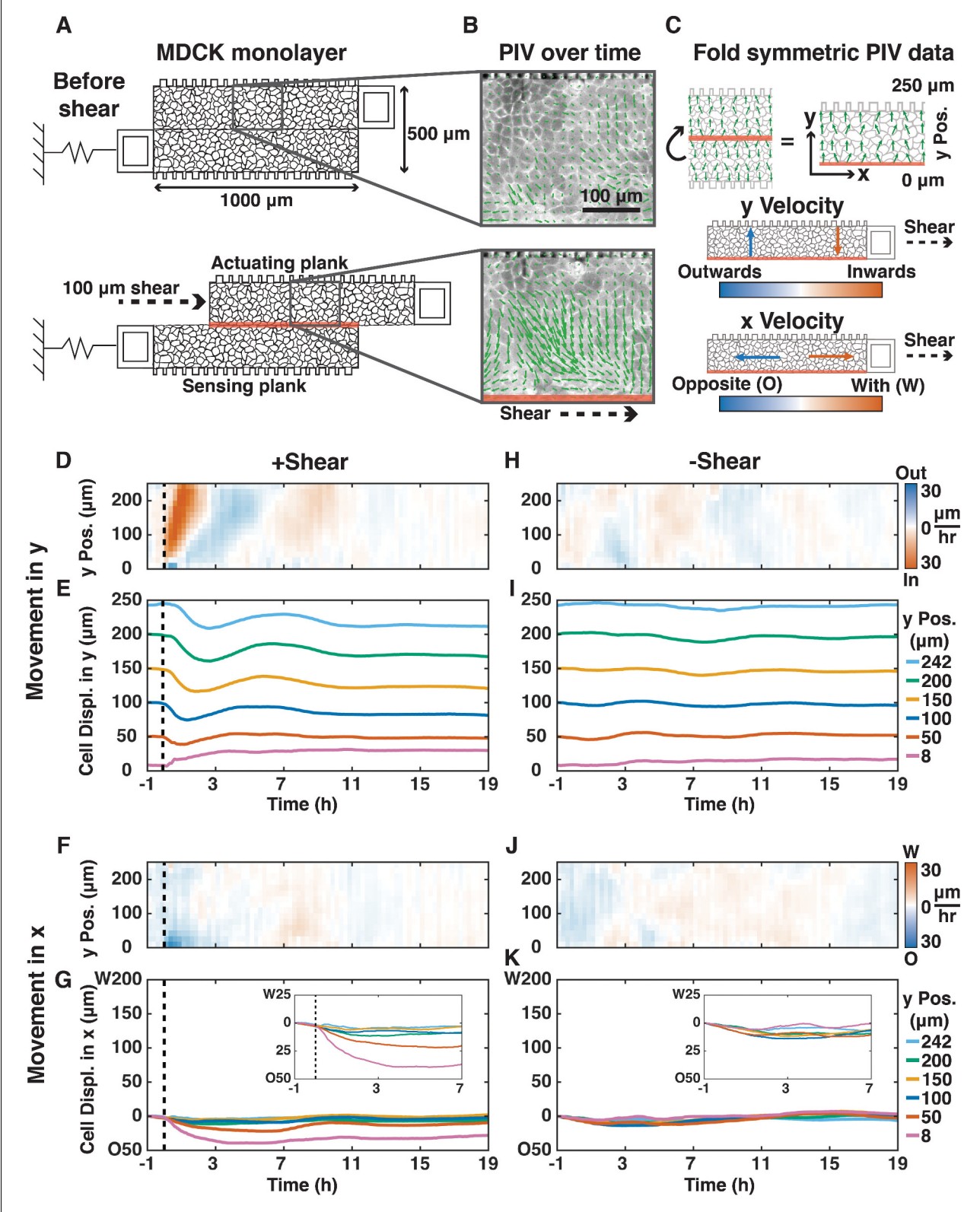

**Figure 1.** Shear induced inward/outward oscillations in cells in the y-direction and cell movements opposite to the shear in the x-direction. (**A**) Shear (100 μm) was applied to a MDCK monolayer adhering to the device planks (Supplementary Materials). (**B**) PIV was used to quantify the x- and y-direction velocities (green arrows) of MDCK cells expressing E-cadherin:DsRed over time. (**C**) Symmetric PIV data were averaged by 'folding' over the shear-plane. The color map displays the speeds of cell movement in the outward/inward (blue/red) y-direction relative to the shear-plane (0 μm y Pos.),

*Figure 1 continued on next page*

*Figure 1 continued*

or in the opposite/with x-direction relative to the shear direction (blue/red, 30 µm/h). (D, F, H, J) y- (D and H) and x-velocity (F and J) kymographs from three independent experiments with 15 min binning of three 5 min PIV data of cell movements with (D and F, dashed black line) or without (H and J) shear over 20 h. (E, G, I, K) y- (E and I) and x-direction (G and K) cell movements based on numerical integration of y- and x-velocity kymographs over time, respectively, at positions 8, 50, 100, 150, 200, and 242 µm from the shear-plane (*Figure 1—figure supplement 7*). Insets provide greater spatial resolution of movement in the deformation zone (G and K, insets).

DOI: https://doi.org/10.7554/eLife.39640.002

The following figure supplements are available for figure 1:

**Figure supplement 1.** A silicon device for force sensing and the application of in-plane shear to a cell monolayer.
DOI: https://doi.org/10.7554/eLife.39640.003

**Figure supplement 2.** MDCK E-cadherin:DsRed cell monolayers before and after shear plus tension show the monolayer remains intact.
DOI: https://doi.org/10.7554/eLife.39640.004

**Figure supplement 3.** MDCK E-cadherin:DsRed cell monolayers before and after shear show deformation localized to the shear plane.
DOI: https://doi.org/10.7554/eLife.39640.005

**Figure supplement 4.** Cell orientation, eccentricity, area, density, and perimeter over time after shear did not match the periodicity of y-direction oscillations.
DOI: https://doi.org/10.7554/eLife.39640.007

**Figure supplement 5.** High-magnification of MDCK E-cadherin:DsRed cells at the shear plane shows no change in E-cadherin asymmetry or recruitment upon shear.
DOI: https://doi.org/10.7554/eLife.39640.009

**Figure supplement 6.** Density of MDCK cells without shear was similar to their density with shear.
DOI: https://doi.org/10.7554/eLife.39640.008

**Figure supplement 7.** Velocity kymographs and cell-displacement plots generated from PIV data.
DOI: https://doi.org/10.7554/eLife.39640.006

within ~1 hr, and stopped after ~3 hr. In this deformation zone, y-direction cell movements were small, outward, and short-lived. Cell oscillations in the y-direction, but not the x-direction, also occurred spontaneously without shear (*Figure 1H–K*; *Video 3*), at a cell velocity 3-4x slower than in the presence of applied shear (9 vs. 30 µm/h). It has been previously reported that MDCK cells plated on circular micro-patterns with 500 µm diameter, the same as the width of the two planks in our device, oscillated radially, but the origin and mechanism of propagation of these oscillations were unknown (*Deforet et al., 2014*; *Kocgozlu et al., 2016*).

To examine whether there were changes in individual cell morphology after in-plane shear, we measured cell orientation, eccentricity, area, density, and perimeter over time. We found no changes in cell morphology or periodicity that matched long-term y-direction collective oscillations (*Figure 1—figure supplement 4*). Thus, inward/outward y-direction oscillations within the epithelium primarily represented collective cell movements rather than changes in individual cell morphology. While the deformation zone (2–3 cell layers) adjacent to the shear-plane exhibited cell-shape changes and horizontal displacement in the x-direction, this physical shear deformation did not occur (*Figure 1—figure supplement 3*, *Figure 1—figure supplement 5*) nor propagate to the rest of the monolayer (*Figure 1—figure supplement 4A,B*). Finally, there was no change in the localization of E-cadherin at cell-cell adhesions of migrating cells due to shear (*Figure 1—figure supplement 5*).

We sought to connect these experimental data to the mechanical properties of the monolayer by using the known stiffness and displacement of the on-chip sensing plank to extract force as a function of time after shear (*Figure 1—figure supplement 1*). The measured position of the sensing plank was used to calculate the force experienced by the epithelium from images taken at 30 s intervals from 5 min before to 30 min after shear (*Figure 2A*). The measured force peaked ($F_{MAX}$) immediately after shear, and then relaxed with an exponential decay characteristic of a viscoelastic material, where the time constant, $\tau$, is

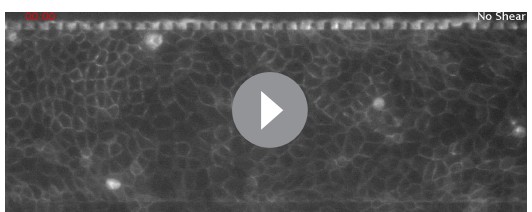

**Video 1.** MDCK E-cadherin:DsRed monolayer imaged 1 hr before shear and 19 hr after shear.
DOI: https://doi.org/10.7554/eLife.39640.010

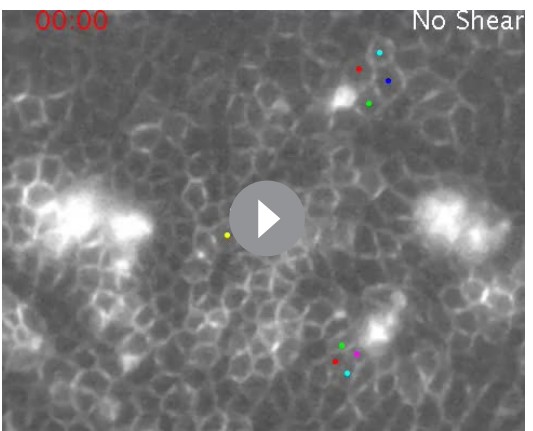

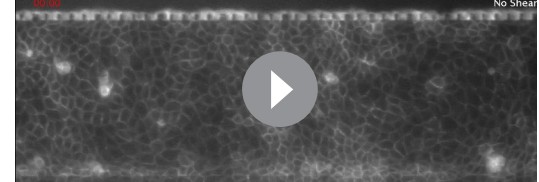

**Video 3.** MDCK E-cadherin:DsRed monolayer imaged for 20 hr without shear.
DOI: https://doi.org/10.7554/eLife.39640.012

**Video 2.** Cell tracking within MDCK E-cadherin:DsRed monolayer reveals cells retain their neighbors before and after shear. Representative image series of the middle third bottom plank of a shear MDCK E-cadherin:DsRed monolayer. Cells were tracked in the shear zone, middle and at the edge of the monolayer using ImageJ's Tracking plugin. Sets of 4 cells were tracked for 1 h before shear and 6.75 h after shear.
DOI: https://doi.org/10.7554/eLife.39640.011

defined as the time to decay by 63% as the epithelium relaxed and remodeled (*Figure 2A*).

Spring and dashpot elements have been used to model the elastic and viscous components of viscoelastic materials, respectively (*Micoulet et al., 2005*; *Mukundan et al., 2013*). While these elements are largely sufficient for modeling normal viscoelastic materials and living materials over short time scales, they do not capture the long-term behavior of active, living materials. The MDCK epithelium in our study behaved as a damped oscillator over tens of hours (*Figure 1D,E,H and I*). Therefore, we inferred that an additional mechanical element was required to capture the time required for mechanical signals to pass through the epithelium, and for the epithelium to generate forces and respond. The addition of a mechanical signal storage and relay element called an *inerter*, placed in parallel with the viscoelastic components captured these damped oscillations over tens of hours (*Figure 2B*). The size of the inerter is inversely proportional to the rate of mechanical signal transduction within the epithelium (80–100 µm/h in *Figure 1D*). The addition of an inerter to viscoelastic elements has been suggested before (*Popović et al., 2017*), but the configuration presented previously did not support the observed oscillations in this study. In our model, $F_{INT}$ represents the intrinsic shear force spontaneously generated by an epithelium to maintain an external force balance (*Prost et al., 2015*), and $F_{EXT}$ is the extrinsically applied shear force. Fitting our model with and without $F_{EXT}$ (+Shear, *Figure 1D*; -Shear, *Figure 1H*) to observed cell movements, we estimate that the $F_{INT}$ required to restore a force balance is 4X smaller than $F_{EXT}$ (*Figure 2—figure supplement 1*).

To interrogate long-term collective epithelial behavior (*Figure 1D and H*), we averaged the y-direction cell velocities from kymographs for the +Shear (*Figure 1D*) and -Shear (*Figure 1H*) conditions. There was no statistically significant difference in the period or damping rate of oscillations (*Figure 2—figure supplement 2C*, p<0.05), but the amplitude of oscillations was higher with shear than without shear (*Figure 2—figure supplement 2C*, p<0.05). We used the MATLAB Simulink/Simscape environment to simulate the behavior of the mechanical models presented in *Figure 2B* (*Figure 2—figure supplement 1*; Materials and methods). The simulation (*Figure 2C,D*, Simulation) captured the observed long-term oscillations and damping after applied shear (*Figure 2C,D*, Experiment). Our models and simulations are in the y-direction , where they account for the shear forces as an inward step input of force (*Janmey et al., 2007*).

The role of actomyosin contraction in shear-induced cell movements in the x- and y-directions was tested by adding the myosin II inhibitor blebbistatin (50 µM; (*Straight et al., 2003*)) 15 min before shear for 1 hr (*Figure 3A–G*; *Video 4*) or 1.5 hr after shear for 1 hr (*Figure 3—figure supplement 1*; *Video 5*). Addition of blebbistatin before (*Figure 3A,B*) or after shear (*Figure 3—figure supplement 1A,B*) inhibited all y-direction oscillatory cell movements, even after washout. Consistent with deformation of a passive material, x-direction cell movements in the 2–3 cell layer deformation zone still occurred after treatment with blebbistatin (*Figure 3C,D*), although the amount of cell movement was less than in the +Shear condition (*Figure 1F,G*). Sensing data revealed no statistical

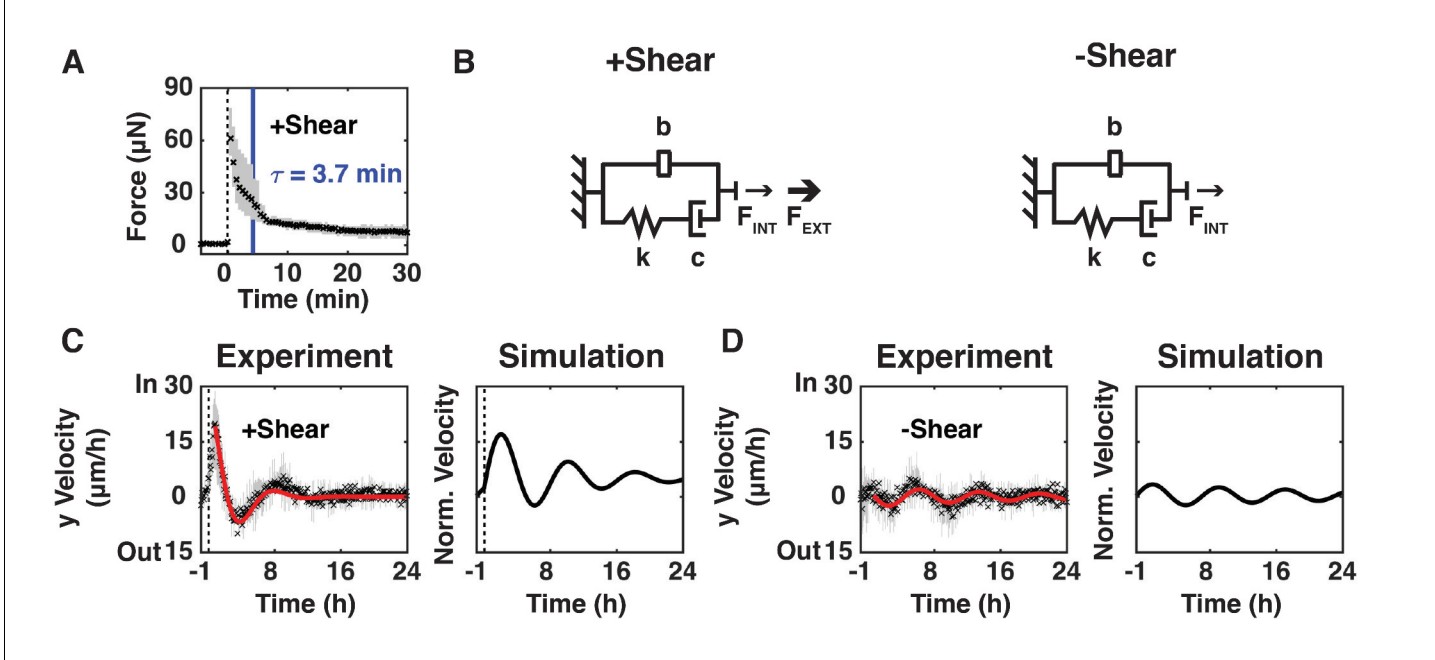

**Figure 2.** Epithelium oscillated over tens of hours to dissipate force imbalance generated by intrinsic or extrinsic shear forces. (A) The measured resistive force of the epithelium over 30 min in response to 100 μm shear displacement for +Shear relaxed with an exponential decay characteristic of a viscoelastic material (dashed line, time of shear; solid blue bar, 63.2% decay time constant τ; *Figure 2—figure supplement 1*). (B) The model captures mechanical properties of the epithelium as stiffness (k) and damping (c) elements in series with each other and in parallel with a mechanical signal storage and relay element (b). $F_{INT}$ represents the intrinsic shear force of the epithelium and $F_{EXT}$ represents the extrinsically applied shear force. (C, D) (Experiment) Unbinned (5 min) y-velocity kymographs for + Shear (*Figure 1D*) and -Shear (*Figure 1H*) were averaged in the y-direction to obtain the overall y-direction cell velocities at each time point with (C), dashed black line) or without (D) shear. The range (gray bars) and mean (black cross) of the data come from three independent experiments at each time point. Red lines represent the best fit of an exponentially decaying sinusoidal function (*Figure 2—figure supplement 2B*) to the average y-direction cell velocity for +Shear and -Shear conditions. (Simulation) Normalized output of the MATLAB Simulink/Simscape simulation for the mechanical model (*Figure 2—figure supplement 1*).

DOI: https://doi.org/10.7554/eLife.39640.013

The following figure supplements are available for figure 2:

**Figure supplement 1.** Simulation of mechanics of local cell deformation and global cell behavior.

DOI: https://doi.org/10.7554/eLife.39640.015

**Figure supplement 2.** Statistical analysis of $F_{MAX}$, τ, and components of exponentially damped sinusoidal function from all experimental conditions.

DOI: https://doi.org/10.7554/eLife.39640.014

difference in $F_{MAX}$ but a much faster relaxation time (τ = 1.3 min) compared to the +Shear condition (*Figure 2—figure supplement 2A*, τ = 3.7 min, p<0.01). In our mechanical model, this faster force relaxation time corresponds to decreased damping (c), and the inverse relationship between the size of the inerter and the rate of mechanical signal propagation leads to a much larger inerter value (b) than in the +Shear condition (*Figure 3F* vs. *Figure 2B*). As expected, the vertical average of y-velocity kymographs (*Figure 3A*) did not exhibit any oscillatory behavior (*Figure 3G*, Experiment). The simulation (*Figure 3G*, Simulation) captured the absence of oscillations (*Figure 3G*, Experiment). These results show that temporary loss of actomyosin contraction, before or after shear, decreases damping, suppresses collective oscillatory behavior, and prevents the propagation of cell movements away from the shear-plane.

Actomyosin contraction is coordinated between cells via linkage to the E-cadherin cell-cell adhesion complex (*Yonemura et al., 2010*). To distinguish the cytoplasmic actomyosin-anchoring role of E-cadherin from its extracellular, tension-dependent *trans*-cell adhesion role, we used MDCK cells expressing an E-cadherin extracellular domain mutant (T151 cells [*Troxell et al., 2000*]). E-cadherin in T151 cells has a truncated, nonfunctional extracellular domain, but an intact plasma membrane-tethered cytosolic tail that binds actin through catenins (*Troxell et al., 2000*). Importantly, expression of T151 E-cadherin results in the down-regulation of endogenous E-cadherin, but other cell-cell

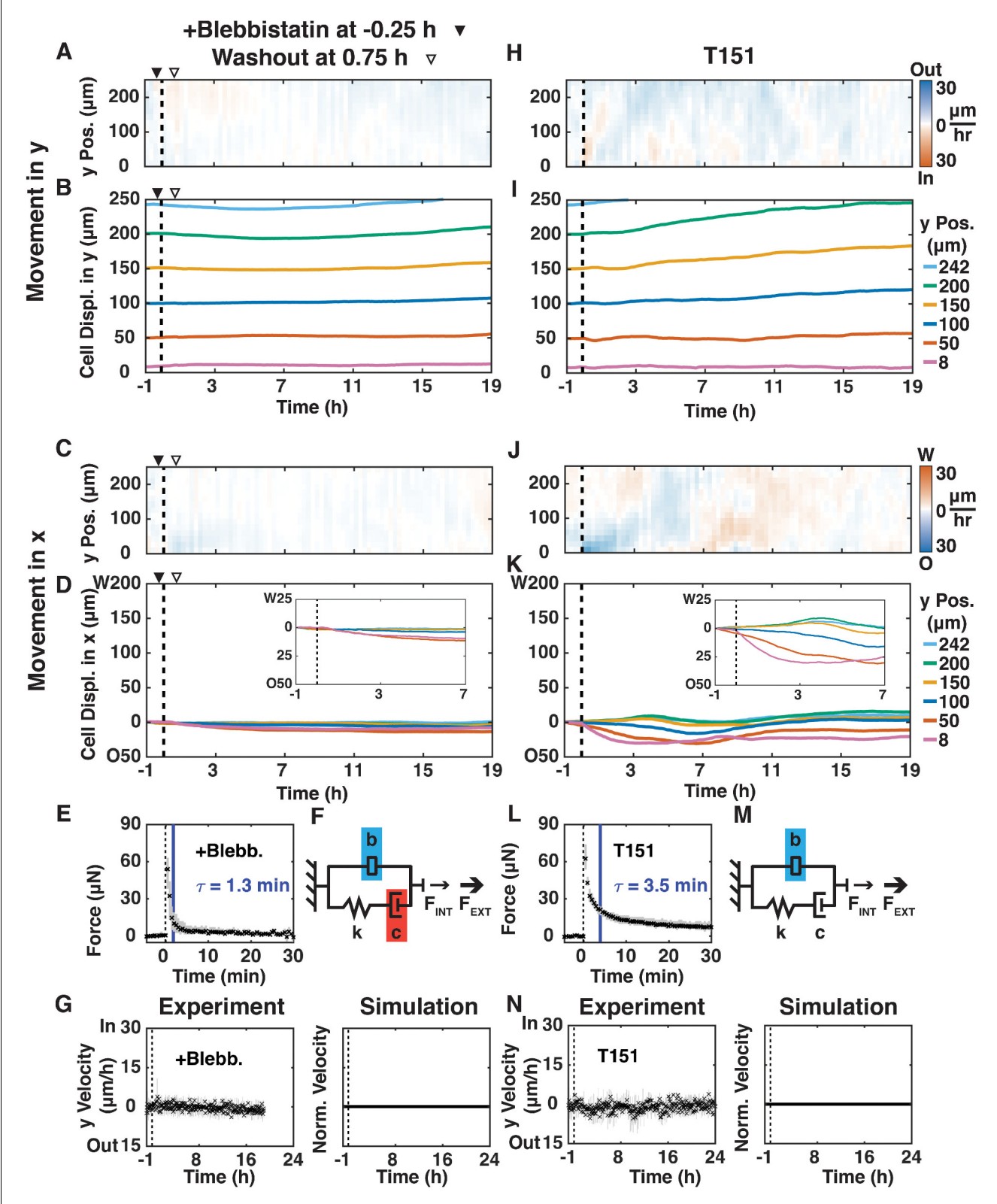

**Figure 3.** Disrupting actomyosin contraction with blebbistatin or loss of E-cadherin-mediated extracellular adhesion disrupted shear-induced y-direction cell oscillations. (A, C, H, J) y- (A, H) and x-velocity (C, J) kymographs from three independent experiments with 15 min binning of three 5 min PIV data of movements of MDCK E-cadherin:DsRed cells treated with 50 μM blebbistatin (+Blebb) (A, C) or of MDCK T151 cells (H, J) and sheared (dashed black line) over 20 hr. (A–D) Blebbistatin was added 15 min before shear (filled arrowhead) and washout one hour later (empty arrowhead). (B,

*Figure 3 continued on next page*

*Figure 3 continued*

D, I, K) y- (**B and I**) and x-direction (**D and K**) cell movements at positions 8, 50, 100, 150, 200, and 242 μm from the shear-plane for +Blebb MDCK cells (**B, D**) and T151 cells (**J, K**), with insets providing greater spatial resolution of passive movement in the deformation zone (**D and K**, insets). (**E, L**) Measured resistive force of the epithelium over 30 min in response to 100 μm shear displacement for +Blebb MDCK cells (**E**) and T151 cells (**L**) (dashed line, time of shear; solid blue bar, 63.2% decay time constant τ; *Figure 2—figure supplement 2*). (**F, M**) Values of the mechanical model elements presented in *Figure 2B* (+Shear) were increased (blue) or decreased (red) to capture the mechanics and movement of the epithelium. (**G, N**) (Experiment) Unbinned (5 min) y-velocity kymographs for +Blebb MDCK cells (**G**) and T151 cells (**N**) were averaged in the y-direction to obtain the overall y-direction cell velocities at each time point with shear (dashed black line). The range (gray bars) and mean (black crosses) of three independent experiments at each time point (**E, G, L, N**) are shown. (Simulation) Normalized output of the MATLAB Simulink/Simscape simulation for the mechanical model (*Figure 2—figure supplement 1*). Kymograph assembly, color maps, data binning, and numerical integration over time were as in *Figure 1*.
DOI: https://doi.org/10.7554/eLife.39640.016

The following figure supplements are available for figure 3:

**Figure supplement 1.** Disrupting actomyosin contraction with blebbistatin blocked and prevented recovery of shear-induced y-direction oscillations.
DOI: https://doi.org/10.7554/eLife.39640.017
**Figure supplement 2.** Density of cells with blebbistatin was similar to their density without it.
DOI: https://doi.org/10.7554/eLife.39640.018

junctions maintain monolayer cohesion (*Troxell et al., 2000*). Shear did not induce inward/outward y-direction movements of T151 cells (*Figure 3H,I*; *Video 6*), unlike the +Shear condition in MDCK cells with normal E-cadherin (*Figure 1D,E*). This result indicates that extracellular E-cadherin *trans*-cell adhesion, and not other cell-cell junctions, was specifically required for the propagation of oscillatory y-direction cell movements after shear. Similar to the blebbistatin-treated epithelium of MDCK cells and the +Shear condition, 2–3 layers of T151 cells adjacent to the shear-plane (<50 μm) moved in the x-direction opposite to shear (*Figure 3J,K*), confirming that passive deformation of the monolayer is independent of E-cadherin *trans*-cell adhesion. The sensing data (*Figure 3L*) revealed no statistical difference in $F_{MAX}$ and τ compared to the +Shear condition (*Figure 2—figure supplement 2*). In our mechanical model, similar to the blebbistatin-treated MDCK epithelium (*Figure 3F*), the inverse relationship between the size of the inerter and the rate of signal propagation leads to a much larger inerter value (b) than in the +Shear condition (*Figure 3M*). The vertical average of y-velocity kymographs (*Figure 3H*) also did not exhibit any oscillatory behavior (*Figure 3N*, Experiment), which was also captured by the simulation (*Figure 3N*, Simulation). Taken together, data from T151 cells demonstrate that damping is unaffected by the loss of E-cadherin *trans*-cell binding, but E-cadherin transcellular engagement is required to relay the mechanical signal from the shear plane through the epithelium.

We next examined whether shear-induced cell movements were affected by reducing actin dynamics and increasing actin filament length by adding jasplakinolide (200 nM; (*Bubb et al., 1994*)) 15 min before shear for 1 h (*Figure 4A–G*; *Video 7*); immunofluorescence imaging confirmed that 200 nM jasplakinolide led to the reorganization of long actin filaments in this cell type (*Figure 4—figure supplement 1*). After shear, jasplakinolide treated cells moved inward in the y-direction at a rate (10 μm/h; *Figure 4A*) 3x slower than in the + Shear condition with untreated cells (30 μm/h). This inward movement propagated to the edge of the plank at a rate 8x faster than the average cell

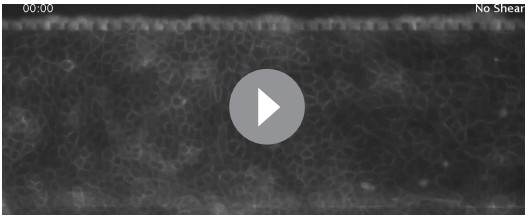

**Video 4.** MDCK E-cadherin:DsRed monolayer imaged for imaged 1 h before shear and 19 h after shear. Blebbistatin was added 15 min prior to shear and washed out 1 h later.
DOI: https://doi.org/10.7554/eLife.39640.019

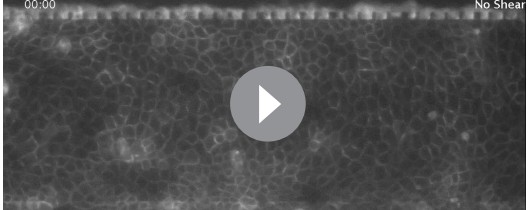

**Video 5.** MDCK E-cadherin:DsRed monolayer imaged for imaged 1 h before shear and 19 h after shear. Blebbistatin was added 90 min after shear and washed out 1 h later.
DOI: https://doi.org/10.7554/eLife.39640.020

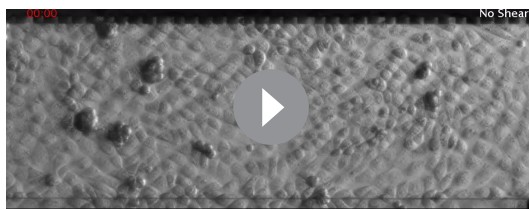

**Video 6.** MDCK T151 monolayer imaged 1 h before shear and 19 h after shear.
DOI: https://doi.org/10.7554/eLife.39640.021

velocity (80 µm/h vs. 10 µm/h), but 3.6x slower than the initial rate in the + Shear condition with untreated cells (290 µm/h, *Figure 1D*). Jasplakinolide treated cells reversed y-direction movements at 7, 9.5, 13, and 17 hr after shear, but without decreasing velocity magnitude (*Figure 4A,B*): the magnitude of oscillations was maintained. After jasplakinolide treatment, cell movement in the x-direction was opposite to the applied shear (*Figure 4C,D*) similar to the +Shear condition with untreated cells (*Figure 1F,G*), although the x-direction movement propagated to cells further from the shear plane. These results indicate that regulation of actin dynamics and actin polymer length is involved in propagating and damping shear-induced oscillatory cell movements in both the x- and y-directions.

The sensing data from the jasplakinolide treatment revealed no statistically significant difference in $F_{MAX}$, but a much slower (*Figure 2—figure supplement 2C*, $p<0.05$) relaxation time ($\tau = 6.2$ min) compared to the +Shear condition with untreated cells ($\tau = 3.7$ min). In our mechanical model, the slower force relaxation time corresponds to an increased damping (c), while the similar rate of mechanical signal propagation (80–100 µm/h) leads to the same inerter value (b) compared to the +Shear condition with untreated cells (*Figure 2B*). The vertical average (*Figure 4G*, Experiment) of y-velocity kymographs (*Figure 4A*) showed that the inward/outward oscillations of jasplakinolide treated cells took much longer ($p<0.05$) to dampen, although with no statistically significant difference in oscillation period compared to the -Shear or +Shear conditions with untreated cells (*Figure 2—figure supplement 2*). Again, the simulation (*Figure 4G*, Simulation) captured the observed long-term oscillations after applied shear (*Figure 4G*, Experiment). Thus, reduced actin dynamics and longer actin polymer length increased damping and slowed the dissipation of the mechanical signal within the monolayer, leading to prolonged oscillations.

## Discussion

Collectively, our results support the idea that local shear affects the epithelium in two spatially and temporally distinct phases, each with different dependencies on the actin cytoskeleton and E-cadherin-mediated cell-cell adhesion. In one phase, x-direction movements and deformations in 2–3 cell layers adjacent to the shear-plane are independent of the organization and contraction of the actin cytoskeleton, and E-cadherin *trans*-cell adhesion. These movements last several minutes, and do not oscillate except when actin dynamics are disrupted. In another phase, y-direction cell movements in the epithelium behave as a damped oscillator, requiring actomyosin contraction, actin filament dynamics, and E-cadherin-mediated cell-cell adhesion; these movements persist for tens of hours.

Based upon our simulations and experimental data, we suggest that transient local deformation of 2–3 cell layers adjacent to the shear-plane generates a mechanical event that is relayed across the epithelium over tens of hours. Cell movements at the shear-plane may induce tension on E-cadherin in cells adjacent to the deformation zone, and tension on E-cadherin increases actomyosin contraction and cell stiffening (*Borghi et al., 2012*; *Buckley et al., 2014*; *le Duc et al., 2010*). Therefore, we suggest that the mechanical event from the deformed cells is transmitted throughout the epithelium via direct coupling of actomyosin contraction and E-cadherin cell-cell adhesion (*Borghi et al., 2012*; *Buckley et al., 2014*; *le Duc et al., 2010*; *Yonemura et al., 2010*). Note that a temporary (1 h) loss of actomyosin contraction blocked long-term (24 h) y-direction cell oscillations even though actomyosin contraction was re-established after blebbistatin washout (+Blebb; *Figure 3A*). These data indicate that with the loss of actomyosin contractility, the generated mechanical signal dissipated, which enabled the epithelium to immediately reach a new external force balance. We suggest that dissipation of the mechanical signal under the -Shear and +Shear conditions occurred through oscillatory cell movements over >15 h, which gradually damped global cell movements as a new external force balance was reached.

The longer force relaxation time constant for cells treated with jasplakinolide showed that damping was significantly increased (*Figure 4E*) because of the longer, stable actin filaments that formed

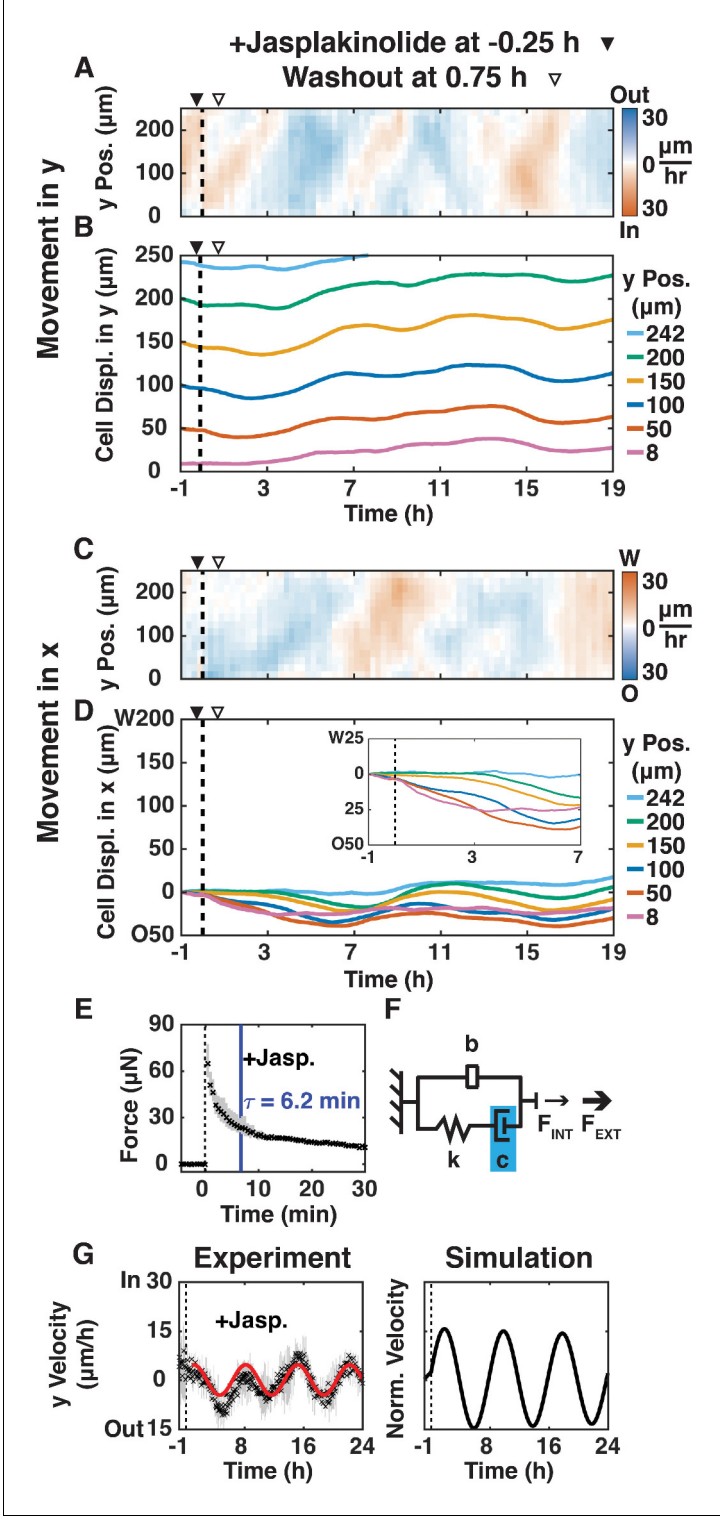

**Figure 4.** Reducing actin dynamics with jasplakinolide prolonged y-direction oscillations. (**A, C**) y- (**A**) and x-velocity (**C**) kymographs from three independent experiments with 15 min binning of three 5 min PIV data of cell movements of MDCK E-cadherin:DsRed cells treated with 200 µM jasplakinolide and sheared (dashed black line) over 20 h. (**A–D**) Jasplakinolide was added 15 min before shear (filled arrowhead) and washout one hour later (empty arrowhead). (**B, D**) y- (**B**) and x-direction (**D**) cell movements at positions 8, 50, 100, 150, 200, and 242 µm from the shear-plane, with inset providing greater spatial resolution (**D**), inset of movement in the deformation zone. (**E**) Measured resistive force of the epithelium over 30 min in response to 100 µm shear displacement

*Figure 4 continued on next page*

*Figure 4 continued*

(dashed line, time of shear; solid blue bar, 63.2% decay time constant τ; *Figure 2—figure supplement 2*) decayed more slowly than + Shear (*Figure 2A*), suggesting higher damping. (F) Values of the mechanical model elements in *Figure 2B* (+Shear) were increased (blue) or decreased (red) to capture the mechanics and movement of the epithelium. (G) (Experiment) Unbinned (5 min) y-velocity kymographs were averaged in y-direction to obtain the overall y-direction cell velocities at each time point with shear (dashed black line). Range (gray bars) and mean (black cross) are shown for three independent experiments at each time point. The red line represents the best fit of an exponentially decaying sinusoidal function (*Figure 2—figure supplement 1B*) to the average y-direction cell velocity (E, G). (Simulation) Normalized output of the MATLAB Simulink/Simscape simulation for the mechanical model (*Figure 2—figure supplement 1*). Kymograph assembly, color maps, data binning, and numerical integration over time are as *Figure 1*.

DOI: https://doi.org/10.7554/eLife.39640.022

The following figure supplements are available for figure 4:

**Figure supplement 1.** High-magnification of MDCK E-cadherin:DsRed cells treated with Jasplakinolide show altered F-actin organization.
DOI: https://doi.org/10.7554/eLife.39640.023

**Figure supplement 2.** Density of cells with jasplakinolide was similar to their density without it.
DOI: https://doi.org/10.7554/eLife.39640.024

---

in the presence of jaspakinolide (*Figure 4—figure supplement 1*). Extrapolating from our experimental data (*Figure 4G*, Experiment) beyond the period of observation indicates that oscillations would have persisted for approximately 80 h after shear. Jasplakinolide stabilizes and increases actin filament length (*Bubb et al., 1994*), which could generate a more viscous actin network that causes the epithelium to act like a material with longer polymeric chains and high damping (*Kim et al., 2016*). This change in physical characteristics may also explain why oscillations occurred in the x- as well as the y- direction after jasplakinolide treatment (*Figure 4A and C*). Thus, normal actin filament dynamics and turnover may be required for oscillation damping and dissipation of the mechanical signal generated by shear, which are required to reach a new force balance.

We can rule out the possibility that changes in cell density over the time of the experiment affected the response of the epithelium to shear. We detected little or no cell extrusion of dead cells (*Video 1*), and the amount of cell division was similar under all experimental conditions with or without shear (*Videos 1* and *3–7*). Furthermore, the cell density was between 6,000–10,000 cells/mm², due to temporary local variations, regardless of the application of shear (*Figure 1—figure supplement 4vs*. *Figure 1—figure supplement 6*), and either in the presence of blebbistatin (*Figure 3—figure supplement 2*) or jasplakinolide (*Figure 4—figure supplement 2*).

Our results suggest a mechanism for the spontaneous oscillation of confined epithelia (*Deforet et al., 2014*; *Kocgozlu et al., 2016*), which are important model systems for embryo development and tumor progression. Spontaneously generated shear forces (*Prost et al., 2015*) in these epithelia produce a normal stress towards the shear plane, (*Janmey et al., 2007*). These researchers posited that their results were relevant to the study of biological tissues and, indeed, may be applicable to our work because of the short time scales associated with reaching peak force (*Figures 2A*, *3E, L* and *4E*). These spontaneously generated shears lead to a force imbalance that is eliminated through the inward/outward oscillation of the epithelium. Here, we reconstituted a similar imbalance by applying a larger external shear force to the epithelium (+Shear), which led to a response similar in type but larger in magnitude to the condition where no external shear force was applied (-Shear). This new understanding was made possible by the ability to apply a localized and quantifiable shear force.

In summary, our novel device and results answer the question of how local shear on cells induces global changes in multicellular

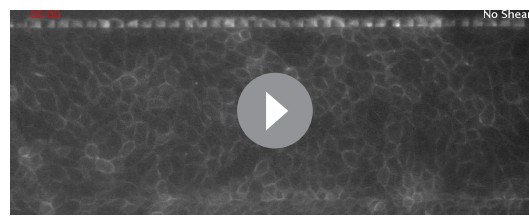

**Video 7.** MDCK E-cadherin:DsRed monolayer imaged for imaged 1 h before shear and 19 h after shear. Jasplakinolide was added 15 min prior to shear and washed out 1 h later.
DOI: https://doi.org/10.7554/eLife.39640.025

organization. Using an intrinsic mechanical signal storage and relay element, we capture the generation of forces by local shear in active, living tissue, and how this signal is propagated and eventually dissipated through oscillatory cell movements. Mechanistically, a force imbalance is generated through local deformation of a few cells by shear. The sensing and restoration of this force imbalance requires tension developed by actomyosin contraction linked to E-cadherin cell-cell adhesion. Our results provide new insights into the mechanical basis of collective cell movement in tissues.

## Materials and methods

### Cell culture and stable cell lines

Madin-Darby Canine Kidney (MDCK) type GII cells stably expressing E-cadherin:DsRed (*Troxell et al., 2000*) or truncated E-cadherin (T151, under control of a doxycycline-repressible promoter, *Troxell et al., 2000*) were cultured at 37°C and 5% $CO_2$ in Dulbecco's Modified Eagle Medium (DMEM) low-glucose (200 mg/L) medium supplemented with 10% fetal bovine serum (FBS) (MDCK Ecad:DsRed) or tetracycline-free FBS (Clonetech, 631106) (T151), 250 µg/mL G418, and 1 g/L sodium bicarbonate, penicillin (30 µg/mL), kanamycin (1 mg/mL), and streptomycin (30 µg/mL). Blebbistatin was added 15 min before shear (50 µM, *Figure 2A–D*) or 1.5 hr after shear (50 µM, *Figure 1—figure supplement 7A–D*) for 1 hr and then washed out with fresh media. Jasplakinolide (0.2 µM, *Figure 2E–H*) was added 15 min before shear for 1 hr and then washed out. All cell lines were tested for mycoplasma prior to experimentation using the LookOut Mycoplasma PCR Detection Kit (Sigma, MP0035).

### Device fabrication

Silicon Microelectromechanical Systems devices were fabricated at the Stanford Nanofabrication Facility from 4-inch silicon wafers (50 µm device layer) using principles and methods presented previously (*Mukundan and Pruitt, 2009*; *Sadeghipour et al., 2017*).

### Shear device preparation

Newly fabricated devices were hydrated in 70% ethanol, rinsed in MilliQ water, and rinsed in 0.01% acetic acid prior to coating with collagen (Corning, 354236, 50 µg/mL in 0.01% acetic acid for 1 hr at room temperature). Devices were re-used after cleaning them with collagenase (2.4 g/mL; Sigma-Aldrich, C0130) for 30 min at 37°C. Devices were placed in concentrated Clorox Regular Bleach for 48 hr, washed in MilliQ water twice for 5 min, incubated in ethanol for 3 hr, rinsed in MilliQ water, and rinsed in 0.01% acetic acid. Cleaned devices were coated with collagen I. Each device was placed in DMEM imaging medium with 50 mM HEPES and 1.8 mM $Ca^{++}$ (Thermofisher, 15630080) supplemented with 10% FBS and 1 g/L sodium bicarbonate and antibiotics as above. A 3D-printed acrylonitrile butadiene styrene holder was used to center and hold the device in a 35 mm polystyrene culture dish.

### Cell seeding

Cell monolayers were dissociated with 0.05% trypsin-EDTA (Life Technologies, 25300–062) for 6 min at 37°C. Cells were resuspended at a concentration of $1.5 \times 10^6$ cells/mL in low-calcium (5 µM $Ca^{++}$) DMEM. A glass capillary tube with an inner diameter of 500 µm was attached to a Rainin p10 micropipette tip. Using a turn-dial micropipette,~2700 cells were pipetted onto the two combined 1000 × 250 µm planks of the device, which was submerged in DMEM imaging medium plus 1.8 mM $Ca^{++}$. The open dish was sealed with mineral oil to prevent evaporation while allowing the insertion and micro-manipulated control of a needle to move the planks. After 1 hr, the medium was agitated to remove non-adherent cells; shear was applied after a further 18 hr.

### Application of in-plane shear

The position of the 'actuating plank' was controlled with a needle attached to a 3-axis micro-manipulator (Newport Corp., UMR8.25), while the force was inferred from the position of the second, spring-loaded 'sensing plank' (*Figure 1—figure supplement 1A*). The actuating plank was displaced 100 µm in the positive x-direction and held for the duration of the experiment (24 hr), which caused in-plane shear deformation at the mid-plane of the monolayer. The rate of shear displacement

was ~20 µm/s. The 100 µm in-plane shear did not rupture individual cells, cell-cell adhesions, or the cell monolayer as a whole.

## Microscopy

Cells were imaged with a Leica DM-RXA2 microscope and a Hamamatsu Orca-R2 camera encased in a black acrylic incubator at 37°C achieved using an Air-Therm ATX. A Leica fluorescent 10x objective was used for all experiments. Images were taken for 45 min at 5 min intervals, and for 15 min at 30 s intervals for a total of 1 hr prior to the application of shear. MDCK E-cadherin:DsRed cells were imaged both in bright field and Texas Red channels to visualize cell shape and sense forces; T151 cells were imaged in only bright field. Images during shear were captured at 200 ms intervals and binned by 2 × 2 for a final resolution of 2.12 pixels/µm. Immediately after shear, images were captured every 30 s for 45 min, and then every 5 min for 23.25 hr.

## Cell segmentation

Fluorescent cell boundaries of MDCK E-cadherin:DsRed cells were used to segment individual cells in the monolayer using custom MATLAB code (*Sadeghipour and Garcia, 2018*) based on techniques presented previously (*Harris et al., 2012*; *Hart et al., 2017*).

## Image processing

Two sets of images were compiled in 5 min interval stacks, 1 hr before shear and after shear for a total of 302 images. Images were divided at the mid-plane to separate the actuating and sensing planks as top and bottom, respectively. Images in each stack were rotated to horizontally align planks and then matched to the first image using the Fiji Template Matching plugin (ImageJ version 2.0.0). The middle third of both the top and bottom planks were cropped for processing to eliminate edge affects of the plank's arms. Image pixel dimensions are 468 × 390 pixels/µm.

## Particle image velocimetry (PIV)

PIV was performed using MATLAB PIVlab 1.41 (The MathWorks). Images were loaded with a 1–2, 2–3, 3–4 style. An Fast Fourier Transform (FFT) window deformation PIV algorithm was used with a first pass interrogation size of 120 pixels with step size 60 pixels, and a second pass interrogation of 48 pixels with a step size of 24 pixels. Window deformation was linear and the method for sub-pixel displacement estimation was Gauss 2 × 3 point. A vector validation with interpolation was applied with these settings: velocity standard deviation filter threshold at 4, local median threshold at 5, epsilon set to 0.1. Velocity and vector data were exported as a. mat file to generate cell velocity kymographs. PIV data from 300 µm x 250 µm (width x height) images were divided into 18 × 15 (width x height) data sets to generate the kymographs. Velocity vectors were averaged horizontally for a final data set of 1 × 15 (width x height). We have provided all source data for each figure as well as the custom Matlab codes (*Sadeghipour and Garcia, 2018*) used to process the data and generate plots. These files can be viewed and downloaded using this link; osf.io/kvu5j.

## Mechanical model

For the mechanical model, stiffness (k) was held constant across experimental conditions, and damping (c) was calculated based on c = kD, from the equation for an exponentially damped sinusoidal function (*Figure 2—figure supplement 2B*). We ensured that the trend in damping values in the mechanical model corresponded with the trend of the τ values from the force relaxation experiments, as τ is a corollary for damping (*Figure 2—figure supplement 1B*). The value of the inerter (b) was inversely related to the rate of propagation within the epithelium, which led to an effectively infinite value where the rate of propagation was zero (+Blebb, *Figure 3A*; T151, *Figure 3H*). When the rate of propagation was not zero (+Shear, *Figure 1D*; -Shear, 1H; and +Jasp, *Figure 4A*), the value of the inerter (b) was chosen to achieve a mechanical system with an intrinsic period (P) of 8 hr, where $b = kP^2/(4\pi^2)$, based on the statistical tests performed and shown in *Figure 2—figure supplement 2*.

## Statistical analysis

Statistical significance of the results was analyzed using a combination of the Kruskal-Wallis and the Mann-Whitney U tests. These non-parametric statistical tests are appropriate when data are not normally distributed. Using data from two experimental conditions in all cases, the Kruskal-Wallis test allowed us to determine whether the results of our different experimental conditions represented different populations. If this test was positive ($p<0.05$), we instead used a pairwise non-parametric Mann-Whitney U test. In *Figure 2—figure supplement 2* we have used red to denote the results using Kruskal-Wallis tests, and black to denote the results using Mann-Whitney U tests.

## Acknowledgments

We thank members of the Nelson and Pruitt laboratories for helpful discussions, as well as nano@-Stanford staff. Work was performed in part in the nano@Stanford labs, which are supported by the National Science Foundation (NSF) as part of the National Nanotechnology Coordinated Infrastructure under award ECCS-1542152. This work was supported by NIH Training Grant T32CM007276 (MAG), NSF and Stanford Graduate Fellowships (ES), the NIH (R35GM118064, WJN), and NSF (CMMI 1662431, BLP and WJN).

## Additional information

### Funding

| Funder | Grant reference number | Author |
|---|---|---|
| National Institutes of Health | T32CM007276 | Miguel A Garcia |
| National Institutes of Health | R35GM118064 | William James Nelson |
| National Science Foundation | CMMI 1662431 | William James Nelson |

The funders had no role in study design, data collection and interpretation, or the decision to submit the work for publication.

### Author contributions

Ehsan Sadeghipour, Conceptualization, Data curation, Software, Formal analysis, Validation, Investigation, Visualization, Methodology, Writing—original draft, Writing—review and editing; Miguel A Garcia, Conceptualization, Data curation, Software, Formal analysis, Validation, Visualization, Methodology, Writing—original draft, Writing—review and editing; William James Nelson, Beth L Pruitt, Conceptualization, Resources, Supervision, Funding acquisition, Investigation, Methodology, Project administration, Writing—review and editing

### Author ORCIDs

Ehsan Sadeghipour http://orcid.org/0000-0002-8686-8315
Miguel A Garcia http://orcid.org/0000-0003-4181-5372
William James Nelson https://orcid.org/0000-0003-3039-3776
Beth L Pruitt http://orcid.org/0000-0002-4861-2124

### Decision letter and Author response

Decision letter https://doi.org/10.7554/eLife.39640.030
Author response https://doi.org/10.7554/eLife.39640.031

## Additional files

### Supplementary files

• Transparent reporting form
DOI: https://doi.org/10.7554/eLife.39640.026

## Data availability

All data generated or analysed during this study are included in the manuscript and supporting files. Source data files and source code have been provided for all figures at: osf.io/kvu5j

The following dataset was generated:

| Author(s) | Year | Dataset title | Dataset URL | Database and Identifier |
|---|---|---|---|---|
| Miguel A Garcia, Ehsan Sadeghipour | 2018 | Source data and source code for all figures | https://osf.io/kvu5j/ | Open Science Framework , kvu5j |

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
