## [Decision Letter]

Thank you for sending your article entitled "Shear-induced Damped Oscillations in an Epithelium Depend on Actomyosin Contraction and E-cadherin Cell Adhesion" for peer review at *eLife*. Your article has been reviewed by Anna Akhmanova as the Senior Editor, a Reviewing Editor and three reviewers.

Summary:

This timely and intriguing manuscript addresses the response of epithelial sheets to shear forces. The authors show that oscillations in cell movements and actin turnover are used to respond to shear forces and return to equilibrium. The authors have conceived innovative and unique technology to explore the phenomenon and use a combination of experiments and theory to analyze the effect of shear on epithelial monolayers. Overall the manuscript is clearly and well written and the experimental data is of good quality and convincing. The reviewers agreed that the manuscript will be broadly of interest to several fields and the data acquired in this study are novel, unique, and are of high quality.

However, the reviewers agreed that are some issues that still need resolved and clarified. These include justification for key experimental parameters, further justification for physical interpretation of the data, key biological controls, and improved statistical analyses.

Essential revisions:

1) The authors base their model, which is at the heart of the study, on the assumption that the viscoelastic properties of the monolayer are not altered by shear (k is constant). Along these lines, in the simulations the oscillations appear more sustained and higher amplitude than in the actual experiments, which could be an indirect indication that the monolayer changes its mechanical response to shear, leading to non-linear dissipation of the oscillations. The authors should quantify the mechanical properties of the monolayer during the response to shear using direct or indirect methods (AFM, traction force etc., depending on what is compatible with their setup) to justify this assumption.

2) The role of E-cadherin is interesting, but the authors statement that the mechanism involves induction of tension on E-cadherin has not been sufficiently explored. The authors could look using IF at the α-18 epitope accessibility and vinculin recruitment as indirect but straight forward readouts of tension, or alternatively use laser ablation to quantify tension over junctions. Is the distribution of E-cadherin changes and does it display anisotropy in respect to the direction of shear?

3) The role of actin remodeling remains vague, and the statement on the role of actin at the end of the abstract appears too strong given that it is based on just the jasplakinolide experiment. Additional experiments should be performed to clarify this and justify the conclusion. Are there changes in the organization of the F-actin cytoskeleton? More x-linking, fiber elongation? What happens to myosin II activity over time and where is the myosin activity localized, is it at junctions. Does myosin activity oscillate?

4) The reviewers had concerns about the large shear forces and how they affect the cells. The authors should analyze whether cell adhesion, shapes or rearrangements are observed in response to shear, the experiments should be repeated at a lower shear strain amplitude to ensure that the data can be interpreted as a purely mechanical phenomena or other explanations of the data should be provided.

Additionally, is cell death/damage induced by the large shears? Evaluation of a live/dead cell stain could alleviate concerns that cell death is occurring. Similarly, the use of non-permeable cell dyes could be used to discern if micro-tears are forming in the cell membranes at the shear plane. Both of these control experiments should be performed to assess the affects of the shear strain on cell health in the MEMS device.

5) The asymmetric cell oscillations (being larger in y than in x, the shear direction) is understandable for the application of shear. However, the explanation for asymmetric oscillations for unsheared cells observed in Figure 1H-K is unclear. In particular, what is the origin of the symmetry breaking cue? Is this due to the different sizes of the x- and y-axes in the MEMS device? If square MEMS devices were used would asymmetric oscillations be observed? The authors claim the oscillations are governed by actin dynamics. Are the actomyosin networks of the cells asymmetrical in terms of their dynamics or distribution even in the absence of shear?

Currently, the observation of the oscillatory motions in both unsheared and shear conditions is interpreted as due to similar mechanisms, but different in amplitude due to differences in the amplitude of the applied shear. However, how the "intrinsic forces" and the applied force have similar directionality is unclear. The similarities of the oscillations in sheared and control conditions could be coincidental due to geometrical constraints of the MEMS device or other issues. In general, further analysis and explanation of the origin of intrinsic oscillations, and their relation to the shear-induced oscillations needs to be added to the manuscript. This could include the experiments suggested in this comment or other experiments deemed more pertinent by the authors.

6) There are a variety of mechanical models and/or physical systems used to explain the various aspects of the epithelia. At various times, the cell layer is considered analogous to a semi-flexible polymer network (work of Janmey), an active gel (work of Prost), and a mechanical circuit. These descriptions are not all compatible. For instance, comparing a cell layer to a semi-flexible polymer network is problematic, as this ignores the active nature of the cells as well as the presence of cell-cell contacts. The comparison to Janmey's work demonstrating the negative normal stresses is particularly problematic because of key microstructural and time scale differences between the systems. The origin of the negative normal stresses in Janmey's work is due to a non-symmetric force extension curve of an individual semi-flexible polymer. If the force-extension curves of cells exhibit a similar asymmetry is not known to this reviewer. Also, the responses observed in Janmey's work occurred within seconds, here the maximal "stresses" in the epithelial occur within hours. This suggests that the observations in this manuscript are due to other mechanisms, likely related to cell migration. If the authors wish to continue using the analogy to Janmey's work, they should demonstrate that cells display the negative normal deformations without active processes to match the conditions in Janmey's work. The easiest way to due to this is through ATP-depletion, but this will likely have substantial biological consequences. The mechanical circuit model, with the inclusion of the inverter, is a novel addition to the field and very interesting. Alternatively, the authors could choose to focus on this model, providing greater explanation of the physical interpretation of the inerter in terms mechanosensitive signaling and remove some the discussion of the polymer physics / active matter.

7) The mechanical circuits have a key time scale of 8 hours. This is a very long time-scale to be associated with actin filament dynamics in living cells, where actin turnover is usually quite high. Jasplak is also known to be a potent anti-proliferation agent. This suggest that cell division could play a key role is dissipate stress in this system. Experiments where cell proliferation is inhibited should be performed to demonstrate or exclude the role of cell division in the dissipation of stresses in the system.

---

## [Author Response]

Essential revisions:1) The authors base their model, which is at the heart of the study, on the assumption that the viscoelastic properties of the monolayer are not altered by shear (k is constant). Along these lines, in the simulations the oscillations appear more sustained and higher amplitude than in the actual experiments, which could be an indirect indication that the monolayer changes its mechanical response to shear, leading to non-linear dissipation of the oscillations. The authors should quantify the mechanical properties of the monolayer during the response to shear using direct or indirect methods (AFM, traction force etc., depending on what is compatible with their setup) to justify this assumption.

It is possible that there are small changes in the stiffness of the monolayer due to shear, but they would not significantly affect our conclusions:

First, any such changes must be small, as a large change would also change the period of oscillations. We do not observe such statistically significant change in the period of oscillations with or without shear (Figure 2 —figure supplement 1C).

Second, the cells on this device are placed on suspended micro-fabricated silicone planks, and the device has been integrated with an upright Leica microscope. None of the common stiffness measurement tools, including those mentioned by the reviewer (e.g., AFM, traction force, etc.) are compatible with this setup.

Finally, we have presented the simplest mechanical model that reasonably matches the mechanical behavior of the cell monolayer by placing all elastic behavior into a single spring, all damping behavior into a single damper, and all mechanical signal storage and relay behavior into a single inerter. The addition of more springs, dampers and inerters in parallel or in series might refine the model but at the expense of increasing complexity. Moreover, such refinements would not alter our conclusion that a storage and relay element (inerter) is required to model monolayer behavior.

2) The role of E-cadherin is interesting, but the authors statement that the mechanism involves induction of tension on E-cadherin has not been sufficiently explored. The authors could look using IF at the α-18 epitope accessibility and vinculin recruitment as indirect but straight forward readouts of tension, or alternatively use laser ablation to quantify tension over junctions. Is the distribution of E-cadherin changes and does it display anisotropy in respect to the direction of shear?

The role of E-cadherin in transmitting tension across cell-cell junctions has been well established (for example, Borghi et al., Yonemura et al., De Rooji et al.). However, instead of indirect methods for assessing whether there was tension on E-cadherin (alpha-18 epitope accessibility, or vinculin recruitment), we wanted to test directly whether tension on E-cadherin specifically was required for the observed oscillatory effects of shear on the cell monolayer. We used a well-characterized MDCK line that expresses a mutant E-cadherin that lacks the extracellular domain (T151), but also lacks endogenous E-cadherin. In T151 cells the tension on E-cadherin is necessarily zero, but cytoplasmic binding partners are retained, and other cell-cell junctions are preserved. Strikingly, we found that shear-induced oscillatory movements of cells completely disappeared. Since other cell-cell junctions are present in T151 cells, we can conclude that shear-induced tension transmission between cells in the monolayer is through E-cadherin. Staining for vinculin, for example, would not provide this critical information. Finally, we examined the localization of E-cadherin-RFP in cells and did not find any change in localization upon or during strain. Finally, the device has been integrated with an upright Leica microscope, and it is not possible to quantify tension over junctions with laser ablation. We will provide high magnification images of the cell-cell junctions as a supplementary figure in a revised manuscript to show no change in the localization of E-cadherin due to shear.

3) The role of actin remodeling remains vague, and the statement on the role of actin at the end of the abstract appears too strong given that it is based on just the jasplakinolide experiment. Additional experiments should be performed to clarify this and justify the conclusion. Are there changes in the organization of the F-actin cytoskeleton? More x-linking, fiber elongation? What happens to myosin II activity over time and where is the myosin activity localized, is it at junctions. Does myosin activity oscillate?

Our conclusion is based on the work of others reporting that jasplakinolide reduces actin depolymerization and increases actin filament length (Bubb et al., 1994). Determining the proper concentration of jasplakinolide is important, so that cell death is not induced but actin dynamics and organization are affected. This can be dependent on cell type and thus we chose 200 nM based on prior work of (Moore et al., 2014). They too used an epithelial cell type treated with 200 nM jasplakinolide and found that this led to a large immobilization of the F-actin network on the lateral sides of cell-cell junctions. We confirmed F-actin reorganization in our cells through IF. Moore et al. proposed that these large F-actin structures could lead to increased tension, as also reported by (Wu et al., 2014). Our work indicates that changes in epithelial mechanics may be altered by changes in the organization (dynamics, polymer length) of the F-actin cytoskeleton. Notably, (Kim et al., 2016) showed that such material would have high damping and, according to our mechanical model, changing that alone simulates the effects observed in sheared jasplakinolide-treated cells. Finally, determining structural changes of F-actin in cells treated with jasplakinolide is complex, particularly in the context of “cross-linking and fiber elongation” suggested by the reviewer, neither of which can be directly measured by imaging methods available on this device.

While we agree that analysis of myosin II localization could be interesting, our goal was to test whether actomyosin activity was required for shear-induced oscillatory movements of cells. Our analysis using blebbistatin shows directly that actomyosin activity is required; localization of myosin II would not enhance or change the conclusions from this direct experiment.

4) The reviewers had concerns about the large shear forces and how they affect the cells. The authors should analyze whether cell adhesion, shapes or rearrangements are observed in response to shear, the experiments should be repeated at a lower shear strain amplitude to ensure that the data can be interpreted as a purely mechanical phenomena or other explanations of the data should be provided.Additionally, is cell death/damage induced by the large shears? Evaluation of a live/dead cell stain could alleviate concerns that cell death is occurring. Similarly, the use of non-permeable cell dyes could be used to discern if micro-tears are forming in the cell membranes at the shear plane. Both of these control experiments should be performed to assess the affects of the shear strain on cell health in the MEMS device.

We reported in Figure S4 that cell shape changes did not match the long-term y-direction collective oscillations. However, we also reported that cells at the shear plane did change shape immediately after shear (Figure 1). We agree that this can be stated more directly in the text, and we can provide images of cells in the shear zone before and after shear at a higher magnification. As for cell adhesion and rearrangements, cells retain their neighbors throughout the experiment and any new cell adhesions are formed normally between divided cells and their neighbors. We will provide this information as a supplementary figure in the revised manuscript.

The cell monolayer does not rupture upon shear and throughout the shear experiments. We will provide images of cells that were sheared and then separated in tension showing a contact monolayer suspended between the two planks as a supplementary figure in the revised manuscript.

MDCK cells have a well-known mode of extruding dead cells. At no time during shear did we see cell extrusion at the shear plane to indicate shear causes cell death. We will provide movies of sheared MDCK monolayers to support this observation.

5) The asymmetric cell oscillations (being larger in y than in x, the shear direction) is understandable for the application of shear. However, the explanation for asymmetric oscillations for unsheared cells observed in Figure 1H-K is unclear. In particular, what is the origin of the symmetry breaking cue? Is this due to the different sizes of the x- and y-axes in the MEMS device? If square MEMS devices were used would asymmetric oscillations be observed? The authors claim the oscillations are governed by actin dynamics. Are the actomyosin networks of the cells asymmetrical in terms of their dynamics or distribution even in the absence of shear?Currently, the observation of the oscillatory motions in both unsheared and shear conditions is interpreted as due to similar mechanisms, but different in amplitude due to differences in the amplitude of the applied shear. However, how the "intrinsic forces" and the applied force have similar directionality is unclear. The similarities of the oscillations in sheared and control conditions could be coincidental due to geometrical constraints of the MEMS device or other issues. In general, further analysis and explanation of the origin of intrinsic oscillations, and their relation to the shear-induced oscillations needs to be added to the manuscript. This could include the experiments suggested in this comment or other experiments deemed more pertinent by the authors.

(Kocgozlu et al., 2016) and (Deforet et al., 2014) used MDCK cells cultured on circular patterns, and found that the cells underwent periods of spontaneous oscillations in direct proportion to the pattern diameter. Significantly, on patterns with a diameter of 500 μm (the same as the combined width of the two planks in our shear device) they observed intrinsic oscillations with a period of ~8 h which we also observed. Neither Kocgozlu et al., Deforet et al., nor we know the origin of spontaneous oscillations in cell monolayers; a speculation is that the oscillations result from the spontaneous shear forces in the monolayer (Prost et al., 2015) where these spontaneous shear forces create tension forces pointing inwards (Janmey et al., 2007) that lead to long-term inward/outward oscillations. We will modify the text in the manuscript to emphasize this point.

6) There are a variety of mechanical models and/or physical systems used to explain the various aspects of the epithelia. At various times, the cell layer is considered analogous to a semi-flexible polymer network (work of Janmey), an active gel (work of Prost), and a mechanical circuit. These descriptions are not all compatible. For instance, comparing a cell layer to a semi-flexible polymer network is problematic, as this ignores the active nature of the cells as well as the presence of cell-cell contacts. The comparison to Janmey's work demonstrating the negative normal stresses is particularly problematic because of key microstructural and time scale differences between the systems. The origin of the negative normal stresses in Janmey's work is due to a non-symmetric force extension curve of an individual semi-flexible polymer. If the force-extension curves of cells exhibit a similar asymmetry is not known to this reviewer. Also, the responses observed in Janmey's work occurred within seconds, here the maximal "stresses" in the epithelial occur within hours. This suggests that the observations in this manuscript are due to other mechanisms, likely related to cell migration. If the authors's wish to continue using the analogy to Janmey's work, they should demonstrate that cells display the negative normal deformations without active processes to match the conditions in Janmey's work. The easiest way to due to this is through ATP-depletion, but this will likely have substantial biological consequences. The mechanical circuit model, with the inclusion of the inverter, is a novel addition to the field and very interesting. Alternatively, the authors could choose to focus on this model, providing greater explanation of the physical interpretation of the inerter in terms mechanosensitive signaling and remove some the discussion of the polymer physics / active matter.

We sought to place our work in the context of previous studies by others. The various models and mechanisms that the reviewer notes have been ways for different groups to study the effects of mechanical perturbations on biological materials – all of which necessarily simplify the physical world. Janmey et al., clearly state that their results are applicable to living biological tissues, including the movement of organelles within cells and the effects of blood pressure on vascular tissues. We have taken care to benefit from Janmey’s work and other papers in very specific instances where the assumptions are applicable, and not in a disorganized or overly broad way. In the context of the work by Prost, we simply make the point that the physics of an active gel leads to a spontaneous shear term. When combined with the work of Janmey, the normal stress resulting from this spontaneous shear is why we observe cell oscillations even when a shear force is not actively applied.

Epithelial migration “average velocity” and not “stress” that peaks 45 minutes after shear. As shown in Figure 2A, Figure 3E, Figure 3L, and Figure 4E, the measured force peaks immediately after shear, and the units of the x-axis on these plots is minutes not hours. We can update the text to further clarify this point. At these short times scales, which are too fast for active processes to take place, the work of Janmey is quite relevant. As we note, some time is required for the mechanical signal about the shear to pass through the epithelium and lead to the peak velocity.

We note that the addition of the inerter helped us understand the long time-scales we observed in our data, and the work from Janmey and Prost gave insights on force imbalances that are caused by shear in the monolayer in the short term. Together our work reveals a unique mechanical mechanism of how tissues relay and dissipate force imbalances, which requires it to be active, and is affected when actin polymer length is increased. We will modify the text to emphasize assumptions and conditions that allowed us to benefit from their conclusions.

Finally, given the volume of medium, the small number of cells and the length of the experiment we believe that ATP depletion is extremely unlikely.

7) The mechanical circuits have a key time scale of 8 hours. This is a very long time-scale to be associated with actin filament dynamics in living cells, where actin turnover is usually quite high. Jasplak is also known to be a potent anti-proliferation agent. This suggest that cell division could play a key role is dissipate stress in this system. Experiments where cell proliferation is inhibited should be performed to demonstrate or exclude the role of cell division in the dissipation of stresses in the system.

We observe cell divisions throughout the course of all experiments, including those with jasplakinolide. Therefore, any differences in the behavior must result from the treatments, which were different, and not cell division, which was present under all conditions.

To exclude the role of cell divisions in the dissipation of stresses in this system we will provide cell density plots from the existing data of drug treated cells. This analysis utilizes cell segmentation and was used for Figure S4.